# Is a Block of the Femoral and Sciatic Nerves an Alternative to Epidural Analgesia in Sheep Undergoing Orthopaedic Hind Limb Surgery? A Prospective, Randomized, Double Blinded Experimental Trial

**DOI:** 10.3390/ani11092567

**Published:** 2021-08-31

**Authors:** Valentina Stenger, Stephan Zeiter, Tim Buchholz, Daniel Arens, Claudia Spadavecchia, Gertraud Schüpbach-Regula, Helene Rohrbach

**Affiliations:** 1AO Research Institute Davos, Clavadelerstrase 8, 7270 Davos Platz, Switzerland; stengervalentina@gmail.com (V.S.); stephan.zeiter@aofoundation.org (S.Z.); tim.buchholz@aofoundation.org (T.B.); daniel.arens@aofoundation.org (D.A.); 2Department of Clinical Veterinary Medicine, Anaesthesia Section, Vetsuisse Faculty Bern, 3012 Bern, Switzerland; claudia.spadavecchia@vetsuisse.unibe.ch; 3Veterinary Public Health Institute of the Vetsuisse Faculty Bern, 3012 Bern, Switzerland; gertraud.schuepbach@vetsuisse.unibe.ch

**Keywords:** peripheral nerve block, sheep, ultrasound, ropivacaine, hind limb surgery

## Abstract

**Simple Summary:**

Many human diseases are not yet fully understood. Tests in animals can support the evaluation of new techniques meant to be applied in humans. Such animal experiments can only be justified with continuous improvements of the analgesic protocols during and after surgery. This study was designed to test the efficacy and feasibility of a technique aiming to desensitize the large nerves of one hind limb in experimental sheep undergoing invasive surgery on one hind limb. This technique was compared to epidural analgesia, a technique known to be effective in alleviating pain but leading to stress due to an inability to move both hind limbs in the early post-operative phase. Nerve blocks of peripheral nerves are widely used in human and veterinary medicine and can improve peri-operative pain therapy. The following study demonstrated that peripheral nerve block provided comparable analgesia to epidural anesthesia. Peripheral nerve blocks of the sciatic and femoral nerves can be used as an alternative to epidural analgesia in experimental sheep.

**Abstract:**

Peripheral nerve blocks are commonly used in human and veterinary medicine. The aim of the study was to compare the analgesic efficacy of a combined block of the femoral and sciatic nerves with an epidural injection of ropivacaine in experimental sheep undergoing orthopaedic hind limb surgery. Twenty-five sheep were assigned to two groups (peripheral nerve block; sciatic and femoral nerves (P); epidural analgesia (E)). In group P 10 mL ropivacaine 0.5% was injected around the sciatic and the femoral nerves under sonographic guidance and 10 mL NaCl 0.9% into the epidural space while in group E 10 mL ropivacaine 0.5% was injected into the epidural space and 10 mL NaCl 0.9% to the sciatic and the femoral nerves. During surgery, heart rate, respiratory rate and mean blood pressure were used as indicators of nociception. In the postoperative phase, nociception was evaluated every hour by use of a purposefully adapted pain score until the animal showed painful sensation at the surgical site. The mean duration of analgesia at the surgical wound was 6 h in group P and 8 h in group E. Mean time to standing was 4 h in group P and 7 h in group E. In conclusion time to standing was significantly shorter in group P while the duration of nociception was comparable in both groups. The peripheral nerve block can be used as an alternative to epidural analgesia in experimental sheep.

## 1. Introduction

Sheep are frequently used as experimental animal models in the context of translational biomedical research [1,2,3,4,5,6,7]. Most orthopaedic trials are performed at the hind limb. They can be very invasive and adequate pain management is essential to avoid suffering. Continuous efforts to optimize the species-specific perioperative care are needed to further reduce stress, discomfort and pain [8].

Perioperative analgesia usually consists of a protocol including systemically administered opioids and nonsteroidal anti-inflammatory drugs [8,9,10,11]. Unfortunately, these protocols might lead to relevant side effects such as gastrointestinal injuries [12,13], hepatotoxicosis [14] and renal dysfunction [15,16,17] and may result in insufficient pain relief [18]. 

Advanced loco-regional anaesthesia techniques represent a valid adjunct to systemic analgesia reducing perioperative pain in humans and various animal species [19,20]. Spinal or epidural injections of analgesic drugs allow effective peri-operative analgesia at the hind limb [18]. Unfortunately, local anaesthetics lead not only to loss of sensation but loss of motor function as well [18,21]. As flight animals, sheep can be severely stressed when unable to stand on their hind limbs while recovering from epidural anaesthesia [22,23,24,25].

The application of peripheral nerve blocks allows reliable perioperative analgesia in humans and animals [26,27,28,29]. With this technique loss of sensation as well as motor function are confined to the operated limb and a combined block of the sciatic and the femoral nerves might lead to profound peri-operative analgesia while standing ability is maintained on three limbs. The most simple technique to apply sciatic–femoral–nerve blocks is the use of anatomical landmarks [19,30]. Nerve stimulators (NS) have been used to improve the success rate of peripheral nerve blocks due to the possibility to localize the nerve through an appropriate motor response. With the aid of ultrasound the nerve, the injection needle, as well as the injected local anaesthetic, can be visualized in real-time. The technique is to be put on a par with the use of nerve stimulators and a good orientation aid due to the safe and accurate representation of all relevant structures [31,32,33,34].

The technique of an ultrasound-guided block of the sciatic and the femoral nerves has been evaluated in sheep cadavers [30]. When the technique was applied in healthy, non-operated sheep, a perineural injection of 10 mL ropivacaine 0.5% to the sciatic nerve led to complete loss of nociception in the target limb [35]. Until now, clinical investigations of this technique with the proposed dose are missing.

The aim of this study was to compare the analgesic efficacy and extent of motor impairment of an ultrasound-guided block of the sciatic–femoral nerves with epidural anaesthesia in sheep undergoing invasive hind limb orthopaedic surgery. We hypothesised that ropivacaine injections to the sciatic and the femoral nerves prior to surgery would lead to improved intra-operative antinociception with longer-lasting postoperative analgesia at the surgical site than epidural injections of the same drug. We assumed that time to standing would be shorter, the general pain and stress levels reduced and the side effects of epidural analgesia as stress due to inability to stand would be avoided when performing the peripheral blocks.

Confirming these assumptions might lead to changes in standard protocols applied in sheep undergoing invasive hind limb surgery in biomedical research and clinical veterinary practice.

## 2. Materials and Methods

The study was designed as a prospective, randomized, double-blinded experimental trial. All sheep were part of an orthopaedic study performed in parallel to this experiment. Two types of invasive hind limb surgeries were applied in the animals involved in this trial: placement of an intramedullary tibia nail (18 animals) and a tibia osteotomy stabilized by a plate (6 animals). The animals receiving an intramedullary nail were grouped separately from the sheep receiving a tibia osteotomy to prevent all animals with tibia osteotomy from ending up in the same group. The invasiveness of both interventions was judged to be comparable. For the intramedullary tibial nail, a surgical approach with more soft tissue trauma was necessary while a larger bone defect was performed in the group undergoing tibia osteotomy.

### 2.1. Animals

Twenty-five adult female Swiss Alpine Sheep, purchased from a local farmer, were included in this study (Figure 5). All animals were clinically examined including an evaluation of neurological deficits and signs of inflammation (haematology including white cell account) prior to the start of the experiment. They were under close clinical surveillance during the entire experimental period. The experiment was approved by the Committee for Animal Experimentation of the Canton Graubuenden, Switzerland (GR 2017_20) and conducted in an AAALAC International accredited institution.

The sheep were housed in groups of 3–4 animals on straw bedding for at least 2 weeks to allow acclimatization. In the stable a 12-h day/night cycle a window allowed constant conditions. Room temperature ranged from 15 to 20 °C, humidity from 30 to 60% and the air exchanged from 10 to 15 times per hour. A maintenance diet consisting of a mixture of straw, hay, silage, maize and salt was fed twice daily while water was available ad libitum in an automatic water drinker. Food but not water was withheld for 36 h prior to anaesthesia. In the animals receiving an intramedullary tibia nail, the housing concept was continued after surgery. The animals receiving a tibia osteotomy were housed together in one stable with individual boxes after surgery. They were placed into purpose-made suspension slings to guarantee restraint and avoid recumbency. Normal standing and limited deambulation were always possible.

### 2.2. Procedure

On the morning of surgery, all animals received an intramuscular (IM) injection of detomidine (0.01 mg kg^−1^; Equisedan; Dr. E. Graeub AG, Bern, Switzerland). After 20 min a catheter was placed into the left or the right jugular vein and general anaesthesia was induced with midazolam (0.2 mg kg^−1^; Midazolam Sintetica 50 mg/10 mL, Sintetica^®^, Mendrisio, Switzerland) and propofol (Propofol 1% MCT; Fresenius Kabi, Bad Homburg, Germany) injected intravenously (IV) and titrated to effect until endotracheal intubation was possible. Sevoflurane (Sevoflurane Baxter 250 mL, Baxter AG, Volketswil, Switzerland, 8152 Opfikon) was delivered in a rebreathing system (Datex Aespire II, Datey Ohmeda S/5 Monitor Anandic Medical Systems AG, Feuerthalen, Switzerland) in 60% oxygen for the maintenance of anaesthesia and an expiratory alveolar concentration of 1.5% was targeted. Before entering the operation theatre carprofen (4 mg kg^−1^; Carprodolor; Virbac, Glattbrugg, Switzerland) was administered IV to all sheep.

Monitoring included cardiovascular (heart rate and arterial blood pressure) and respiratory parameters (oxygen saturation (SpO_2_), respiratory rate (*f*_R_), end tidal carbon dioxide concentration (FE’CO_2_), end tidal sevoflurane concentration (FE’_Sevo_) and nasal temperature. Blood pressure was measured invasively by use of the left or the right auricular artery. Mechanical ventilation was applied to maintain normocapnia directly after onset of anaesthesia.

After placing the animal on the surgical table all sheep received an injection of ropivacaine (0.5%; 10 mL, Ropivacain; Fresenius Kabi, Bad Homburg, Germany) or NaCl (0.9%; 10 mL; B. Braun Medical AG, Sempach, Switzerland) into the epidural space and around each nerve (sciatic and femoral nerve) a volume of 10 mL was injected (20 mL in total) in a double-blinded fashion. The syringes were prepared by a person not involved in the study.

During surgery, all vital parameters were recorded every 5 min. A cristalloid solution was delivered at a rate of 5 mL kg^−1^ hr^−1^ throughout anaesthesia (Ringerlactat, B. Braun Medical AG, Sempach, Switzerland) until recovery. Further measures (e.g., maintenance of body temperature, adaptation of tidal volume, etc.) to maintain all vital parameters in a physiological range were performed according to individual needs.

After completion of the surgery, a modified Robert Jones bandage was applied to the operated limb. The recovery was performed in the stable under intensive surveillance. The duration of anaesthesia as well as the duration of surgery was recorded and evaluated for all animals.

### 2.3. Locoregional Anaesthesia

After induction of anaesthesia, the animal was positioned in sternal recumbency and all 3 injection sites were surgically prepared. A lot was drawn to allocate every animal to either group E (epidural injection) or P (peripheral block). Group E received ropivacaine into the epidural space and NaCl to the peripheral nerves while group P received NaCl into the epidural space but ropivacaine to the peripheral nerves.

The epidural injection was performed using a spinal needle placed into the lumbosacral space (22 Gauge, 2 inch, Perican^®^, B. Braun Melsungen AG, Melsungen, Germany). The correct position of the needle tip was confirmed by the use of the hanging drop method [18,36,37] and the first injection was performed. In the same position, the second dose was injected into the sciatic nerve under real-time visualization by use of a specifically designed injection needle (SonoBlock, 21 G × 100 mm, PAJUNK^®^, Geisingen, Germany) and an ultrasound machine (Sonosite M-Turbo, Siemens, Munich, Germany; Figure 1).

For the third injection, the sheep was placed in lateral recumbency with the limb to be operated uppermost. After sonographic visualization of the femoral nerve and the injection needle the last injection was performed (Figure 2). The person performing the injections was unaware of the content of the syringes. 

The syringe for the epidural injection (injection 1) or the syringes for the peripheral blocks (injection 2: sciatic nerve; injection 3: femoral nerve) contained ropivacaine (Ropivacain 0.5%; 10 mL; Fresenius Kabi, Bad Homburg, Germany) or NaCl (0.9%; 10 mL; B. Braun Medical AG, Melsungen, Germany) depending on the group.

### 2.4. Intra-Operative Evaluation of Nociception

Heart rate (HR), mean arterial pressure (MAP) and respiratory rate (*f*_R_) were recorded for 15 min and considered as baseline values prior to the start of the surgery. Recording of these values was continued during surgery every 5 min and an increase of ≥20% of two of the three parameters (HR; MAP; *f*_R_) was considered indicative of nociception leading to administration of fentanyl (10 mcg kg^−1^ IV) as rescue analgesia. 

### 2.5. Evaluation of Analgesia

Prior to surgery a sheep grimace scale and a multidimensional pain score were applied to all animals once daily for 6 days to allow proof of health of the animals as well as to promote acclimatization to the scoring procedure (Figure 3 and Figure 4) [38]. The multidimensional pain score consisted of the following individual surveys: VAS (Visual Analog Scale, 10 cm horizontal line), general pain score, block evaluation score and pain evaluation score of the operated limb (Figure 4).

Pain evaluation was continued during the post-operative period using the identical multidimensional pain score. The evaluations were started as soon as the animal had regained consciousness and then repeated every 60 min for at least 6 h and continued as long as the locoregional anaesthesia prevented any sensation in the area of the surgical wound. The procedure was performed in a fixed order: evaluation of the grimace scale (Figure 3), VAS, evaluation of HR and RR followed by the multidimensional pain score (Figure 4). The time point of injection of the locoregional anaesthesia was considered as T0 and the last evaluation was performed in the morning after the surgery (Day + 1). 

Methadone (0.1 mg kg^−1^ IV) was administered as rescue analgesia as soon as the general pain score reached ≥6 or VAS ≥ 40 mm. Repeated injections were possible at every evaluation time point. Buprenorphine 0.6 mg (Bupaq, Streuli Pharma AG, Uznach, Switzerland) was administered intramuscularly to all sheep as soon as a sensory reaction could be elicited at the surgical area (≥T360) as well as on the day following surgery (Day 1) after pain evaluation. At the same time point, fentanyl patches (0.002 mg kg^−1^, Fentanyl-Mepha, Mepha Pharma AG, Aesch, Switzerland) were applied to the lateral side of the left or right front limb after shaving and degreasing the skin with alcohol. All scores were completed by the same observer (VS) blinded to the treatment during the entire study period.

### 2.6. Data Collection and Comparisons

Primary outcome measures included intra-operative antinociception, duration of analgesia at the surgical site and mobility (time to standing) during the early post-operative phase. For the evaluation of the quality of intra-operative antinociception, the vital parameters HR (heart rate), RR (respiratory rate) and MAP (mean arterial blood pressure) were compared. Duration of analgesia at the surgical site was assessed using the pain evaluation score of the operated limb (every 60 min for 360 min or until a sensory reaction could be elicited; (Figure 4).

Secondary outcome measures included general signs of pain and stress evaluated by use of the general pain score. The block evaluation score (modified after Bromage [39]) and the pain evaluation of the operated limb were applied after grimace scale, VAS and pain score at the same predefined time points.

### 2.7. Statistical Analysis

Data analysis was performed using statistical software (Sigma Stat, Version 3.5, Systat Software and SAS 9.4, SAS Institute. Inc., Cary, NC, USA). Continuous variables were analyzed with non-parametric statistics due to the relatively low power of the normality test for the sample size in this study. Therefore, all continuous values are described as median [IQR].

Demographic data such as weight and age of the animals were compared between groups by use of the Mann–Whitney Rank Sum Test. Intra-operative data as HR, RR and MAP were first summarized as mean values per animal, then group differences were evaluated by use of the Kruskal–Wallis One Way Analysis of Variance on Ranks. Duration of anaesthesia, duration of surgery as well as effect duration of the locoregional anaesthesia techniques were compared between groups by use of the Mann–Whitney Rank Sum Test.

SGS, reaction to palpation (second part of the pain evaluation of the operated limb), sensation at the coronary band (second part of the block evaluation) and VAS were analyzed with mixed regression models accounting for repeated measures within animals. The time period was included as a within-factor variable while the treatment group was included as a between-factor variable. The interaction between time and group was also included. A logistic model (SAS PROC GENMOD with binomial distribution) was applied to compare SGS over time and between groups. An ordinal model (SAS PROC GENMOD with multinomial distribution) was used to compare sensation at the surgical wound and at the coronary band over time and between groups. A linear model (SAS PROC MIXED) was applied to evaluate the VAS over time and between groups. Only the time period between T3 (3 h after the block/epidural) and T6 (6 h after the block/epidural) was included in these evaluations, because during this period most animals had complete observations. Significance was set at *p* ≤ 0.05.

## 3. Results

Twenty-four sheep completed the study. One sheep of the epidural group had to be replaced due to euthanasia after tibia fracture during recovery. The corresponding data were excluded from data analysis.

The median age of the animals was 3 years (IQR 2–4.6 years; min 2, max 7 years) while the median bodyweight was 69 kg (IQR 62–76 kg; min 51 kg, max 88 kg). No differences between groups regarding age and weight were identified (Table 1). 

The duration of anaesthesia, as well as the duration of surgery, were shorter for the group receiving an intramedullary nail than for the animals receiving a tibia osteotomy (*p* < 0.001) but no difference between groups was detected when the analgesia techniques were compared (Table 2). This means the anaesthetic technique (epidural vs. nerve block) had no influence on the duration of anaesthesia or surgery. Therefore, the type of surgery was not considered as a relevant parameter for the following evaluations.

### 3.1. Intra-Operative Evaluation of Nociception and Rescue Analgesia

The sciatic and the femoral nerves could be identified sonographically in all sheep participating in this study (groups E and P). In all but one sheep of group E, the position of the needle tip in the epidural space could be confirmed by the use of the hanging drop method [37].

During surgery, mean heart rate did not differ between groups (P: 81 (76–94) bpm, E: 85 (79–88) bpm (*p* = 0.807) while mean blood pressure was lower in group E than in group P (E: 68 (64–78) mmHg; P: 99 (84–103) mmHg (*p* = 0.006)). Ventilation was mechanically controlled in all sheep.

One sheep of group P received one fentanyl bolus during surgery 1 h after the block due to an increased heart rate (82 bpm to 105 bpm) and increased mean arterial blood pressure of more than 20% (58 mmHg to 104 mmHg) compared to baseline values. This animal showed clear signs of motor blockade during the post-operative period and the duration of analgesia at the surgical site was 6 h.

### 3.2. Evaluation of Analgesia and Rescue Analgesia

All animals were considered healthy with neither any orthopedic nor neurologic deficits. The multidimensional pain score was 0 in all animals at any evaluation time point prior to surgery.

Duration of analgesia did not differ between groups while time to standing was shorter in group P than in group E (*p* < 0.001, Table 3). Table 3 shows that in the epidural group at the timepoint where the animal could stand (7 h, later postoperative period), the analgesia at the surgical wound was still working. In the peripheral nerve block group, all animals could already stand at this later postoperative period but also had gained sensation at the surgical wound. Furthermore, 5 out of 12 sheep in the peripheral nerve block group showed knuckling and at the time they had gained sensation at the surgical wound but could still not properly use the leg. Interestingly, in the animals of group E sensation first returned at the coronary band, followed by the surgical area but latest at the parasacral area of the pelvic limb. In group P the animals first regained sensation proximally to the wound while sensation at the coronary band returned several hours later.

Three hours after injection of ropivacaine (T3) recovery from general anaesthesia was complete in all animals. Data collection of the SGS and the multidimensional pain score (VAS, general pain score, block evaluation score and pain evaluation score of the operated limb) was performed in all animals between T3 and T6. The mixed regression models did not reveal any group effect for VAS and SGS. When the duration of analgesia at the surgical wound and the duration of loss of sensation at the coronary band were evaluated, the duration of analgesia at the surgical wound was longer in group P despite the proximal-to-distal direction of the return of sensation (Table 4).

Mean heart rates and mean respiratory rates did not differ between groups when the mean heart rates and the mean respiratory rates were compared from T3 until the time point of return of sensation at the surgical site (HR: *p* = 0.386; RR: *p* = 0.34). In two sheep of group P, the pain score was 4 and 6 at T4 and T5, respectively. In one sheep of group E, the pain score increased to 6 at T4 leading to methadone injections. 

### 3.3. Side Effects

Five out of twelve animals of group P showed knuckling postoperatively. These animals were treated with a support bandage at the region of the fetlock. The next morning sensation had returned in all animals and knuckling had resolved overnight in all sheep.

## 4. Discussion

In the present study, the peripheral block of the sciatic and the femoral nerves with ropivacaine 0.5% allowed adequate analgesia during the intra- as well as the postoperative period. Intraoperatively, the antinociceptive effects of this technique were considered to be equal to the epidural administration of the same drug. During the post-operative period, the analgesic effects of both locoregional anaesthesia techniques were adequate as the scores of both groups remained low and only small differences between groups could be detected. This result is consistent with findings in studies in small animal and human medicine [28,29,40]. No correlation between the need for intraoperative rescue analgesia and the need for methadone in the post-operative phase could be detected. Indeed, an incomplete intraoperative block can still provide sufficient post-operative analgesia [41].

Various locoregional anaesthesia techniques were investigated with the aim to improve peri-operative analgesia at the pelvic limb. Studies in human medicine have shown an increased success rate when sonographic guidance was used for the performance of the peripheral nerve blocks [42]. The study from Waag et al. provided a detailed anatomical description of the sciatic and the femoral nerves [30]. In the present study, only one sheep received rescue analgesia intraoperatively, leading to the assumption that the applied technique was adequate and the success rate was nearly 100%.

Ropivacaine is a frequently used local anaesthetic in human and veterinary medicine [18,43,44]. The long-lasting antinociceptive effects are particularly desired in animals undergoing invasive orthopaedic surgeries [45]. When compared to bupivacaine, ropivacaine is slightly less potent but significantly less cardiotoxic and neurotoxic. Due to its positive properties, ropivacaine would also be the drug of choice in a clinical setting [18,46]. The dose of ropivacaine chosen for this study (10 mL of ropivacaine 0.5% as epidural injection and 10 mL ropivacaine 0.5%/nerve) led to a complete loss of nociception in the operated hind limb during surgery and the post-operative phase in nearly all sheep.

The main advantage of the peripheral nerve block was the early mobility of the sheep in the post-operative phase. Mean time to standing was 4 h in group P and 7 h in group E. Since sheep are flight animals they can be severely stressed if they are not able to stand [22]. Early mobility is beneficial due to its positive effects on circulation, respiratory function and gut motility [18]. However, one drawback with the use of peripheral nerve blocks was the frequent occurrence of knuckling postoperatively. Knuckling can lead to cutaneous injuries at the dorsal region of the fetlock joint. A bandage can prevent superficial skin abrasions in the region of the fetlock joint. Another drawback of the peripheral nerve block compared to the epidural injection is the need for more sophisticated equipment as an ultrasound machine and specifically designed injection needles.

According to the literature, a major side effect occurring after epidural injection of local anaesthetics is a drop in blood pressure mediated by a sympathetic blockade and segmental vasodilation [47]. In this study, the mean arterial blood pressure was significantly lower in the epidural group than in the nerve block group. Ventilation was mechanically controlled in all sheep. This was one reason why this parameter was very stable and no difference between groups could be detected. Therefore, this value has no significance in relation to nociception.

The course of the return of sensitivity was exactly the opposite in the sheep receiving an epidural injection to the ones with peripheral nerve blocks. In the peripheral nerve block group, limb sensitivity returned from proximal to distal. Nonetheless, the duration of analgesia at the surgical site was comparable for both techniques. One explanation for the resolution of the peripheral nerve block from proximal to distal is the anatomical structure of the peripheral nerves. Nerve fibres in the centre bundle of the nerve trunk innervate the sensory fibres in the distal limb. The local anaesthetics might first be absorbed in the surrounding tissue of the nerve trunk and at the latest in the core of the nerve trunk. Therefore, the loss of sensation might remain the longest in the distal structures of the limb [48].

On the other hand, a dermatome is a cutaneous area supplied by a single spinal nerve root with cell bodies located in dorsal root ganglia. The spinal nerve roots are distributed to structures according to their associations with spinal cord segments [49]. In dogs, the dermatomes of the distal hind limb are innervated by sacral spinal nerves whereas the croup dermatomes originate from L1–L5 [50]. In this study, the bevel of the spinal needle was inserted in rostral direction forcing the drug to spread in cranial direction. When new methylene blue was administered epidurally to the lumbosacral space in goats, the epidural space up to L3–L4 was coloured [51]. This could explain the long-lasting effect on the skin sensitivity at the croup.

The time course of the return of sensation allows a clear prediction of the duration of post-operative analgesia for a specific limb segment and can be used to select the most appropriate technique. An epidural injection might be more suitable for surgical intervention at the upper limb while a peripheral nerve block might be applied prior to lower limb surgery.

One limitation of the study was the difficulty of recording stress and pain levels in the animals. Signs of pain and stress can be very subtle and hidden at any moment of insecurity. A structured score sheet was applied to allow objective evaluation which was performed by an experienced person blinded to the groups but familiar with the sheep due to the pre-operative scoring training. Any entrance of people non-familiar with this group of sheep induced turbulences. This showed the importance of being familiar with the sheep and the need for training before reliable data can be collected. The correlation between SGS and general pain score was only moderate and lower than expected. This might be based on false high values for the SGS in the early postoperative phase. The sheep were reduced in their general behaviour caused by the general anaesthesia and observations such as orbital tightening and low ear position which are considered to be signs of pain in the SGS erroneously resulted in a high SGS although the animal was probably not in pain.

The absence of a control group without locoregional anaesthesia could be seen as another limitation of this study. However, this group was not included due to ethical reasons. The long-term effects of both techniques were not investigated in this study. In the present study, only animals that were part of an independent orthopaedic study were used to avoid euthanasia of any extra animals. Thus, the number of animals was limited.

## 5. Conclusions

In this study, we demonstrated the feasibility of the application of a peripheral nerve block of the sciatic and the femoral nerves in sheep undergoing invasive hind limb surgery. Intraoperatively, antinociception was not improved with the application of peripheral nerve blocks compared to epidural anaesthesia.

In the early postoperative period, the animals gained the ability to stand more rapidly after a peripheral nerve block. Despite the anatomically opposite direction of the return of sensation, the duration of analgesia at the surgical site was comparable for the two techniques.

In the later postoperative period, a benefit of the epidural analgesia was seen as the sheep were able to walk on all limbs while not yet feeling the surgical wound at the tibia. The anatomical localization of the surgical intervention might guide the choice of the locoregional analgesia method: an epidural injection might be beneficial for proximal hind limb surgery while a peripheral nerve block might be the first choice for distal hind limb surgery.

## Figures and Tables

**Figure 1 animals-11-02567-f001:**
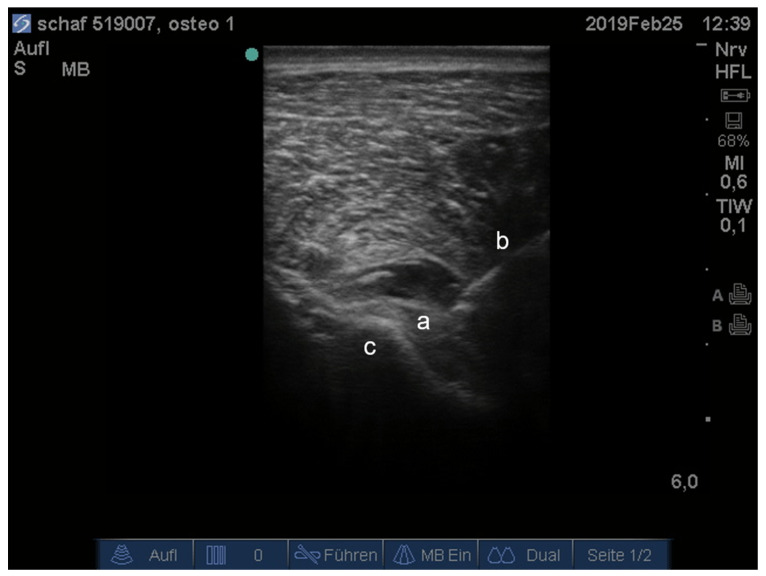
Ultrasonographic image of the right sciatic nerve (a), needle (b), ilium (c).

**Figure 2 animals-11-02567-f002:**
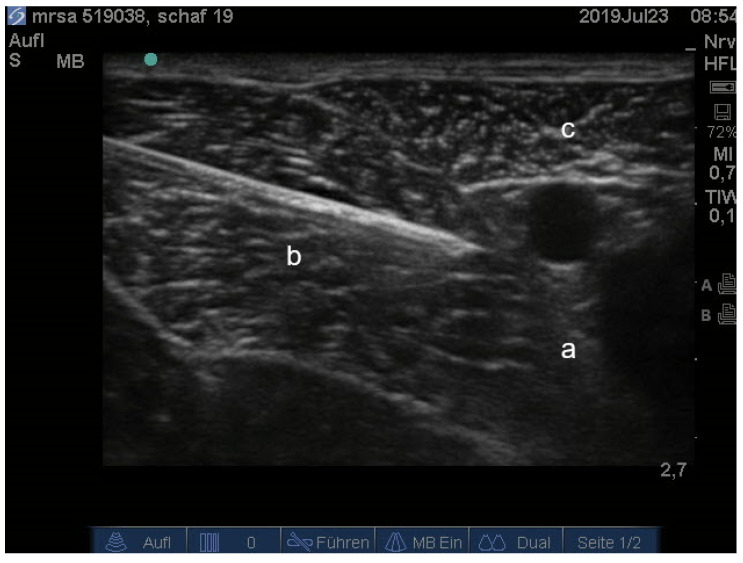
Ultrasonographic image of the left femoral nerve, femoral artery and vein (a), needle (b) and sartorius muscle (c).

**Figure 3 animals-11-02567-f003:**
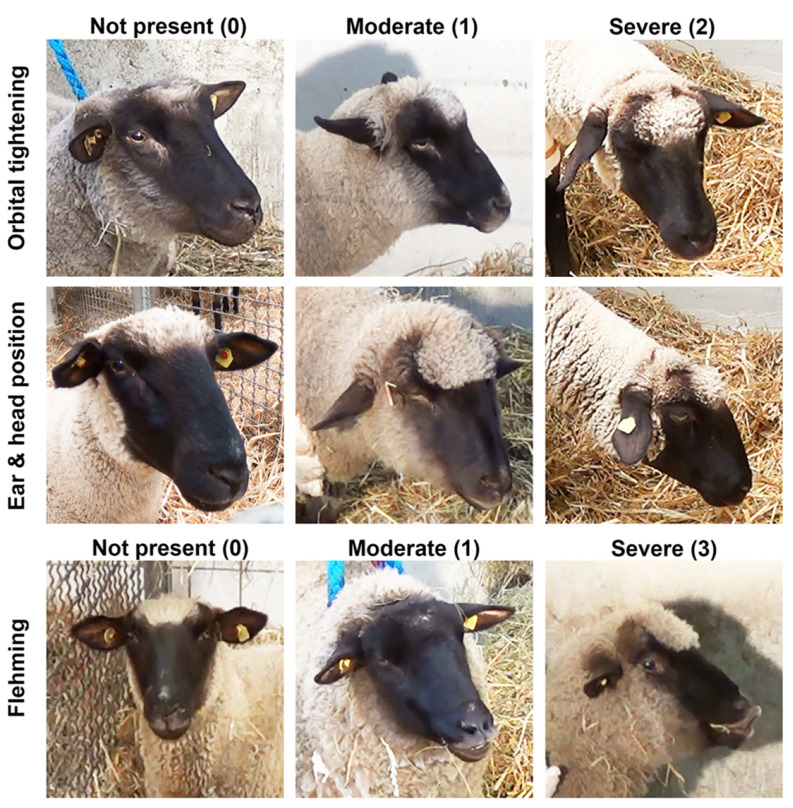
Action Units of the Sheep grimace scale (SGS) [38].

**Figure 4 animals-11-02567-f004:**
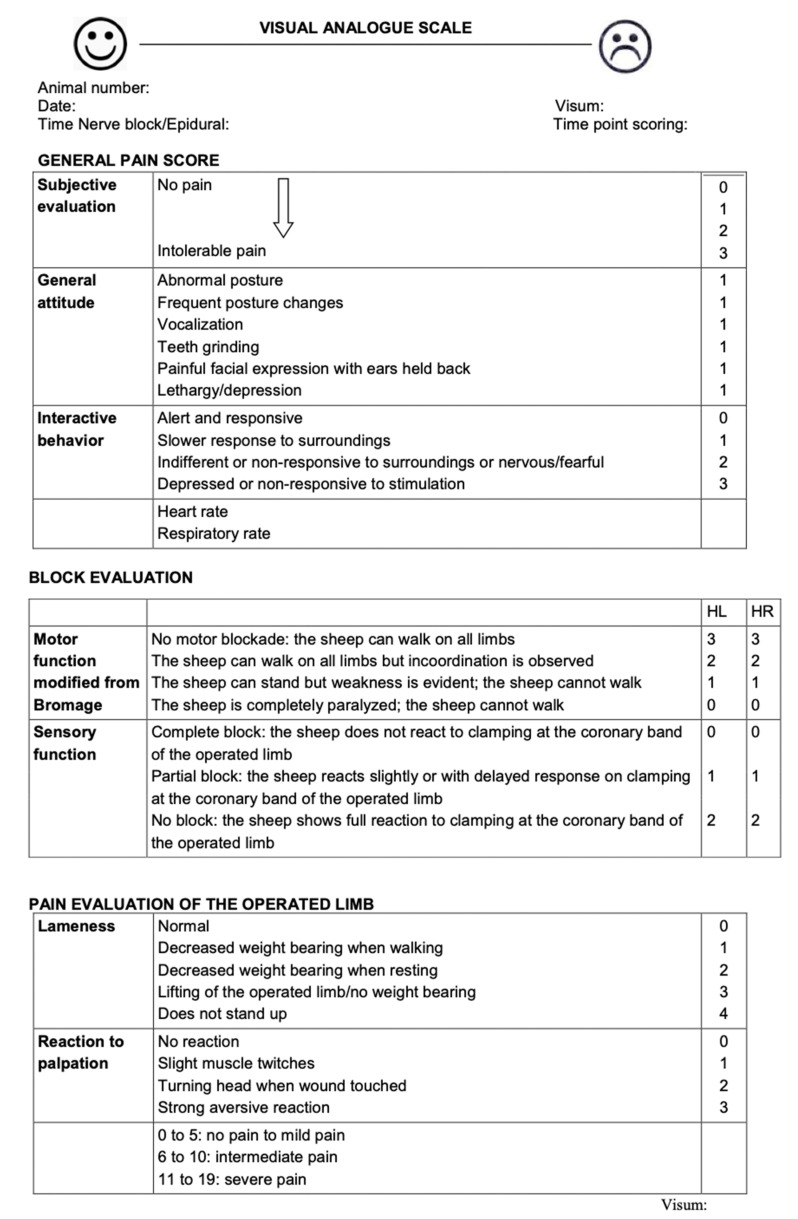
Multidimensional Pain Score consisting of the following individual surveys: VAS (Visual Analog Scale, 10 cm horizontal line), general pain score, block evaluation score and pain evaluation score of the operated limb.

**Figure 5 animals-11-02567-f005:**
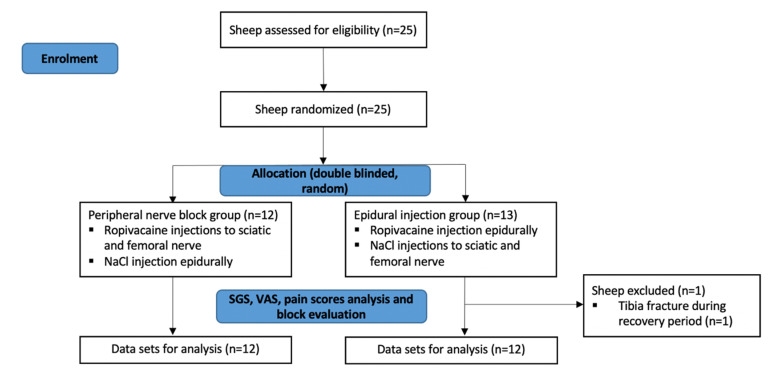
Flow diagram of sheep undergoing hind limp surgery eligible for the study in the time frame of February 2019 to August 2019. The diagram was adapted from CONSORT 2010 (transparent reporting of trials).

**Table 1 animals-11-02567-t001:** Median age and bodyweight did not differ between groups. IQR: interquartile range.

	Peripheral Block	Epidural	*p*-Value
Age (years)	2.75 (IQR 2–3.75)	3 (IQR 2–5.25)	0.597
Weight (kg)	70 (IQR 64–77.75)	68 (IQR 60.9–72)	0.414

**Table 2 animals-11-02567-t002:** Median duration of procedures in sheep undergoing tibia osteotomy or intramedullary tibial nail implantation. No differences could be detected when the durations of the two procedures were compared.

Duration (min)	Peripheral Block	Epidural	*p*-Value
Anaesthesia both types of surgery	153 (130–176)	155 (133–181)	0.9
Surgery both types of surgery	58 (50–84.5)	50 (45–68)	0.3
Anaesthesia tibia osteotomy group	210 (185–225)	232 (212–234)	0.4
Anaesthesia intramedullary nail	148 (125–158)	135 (123–156)	0.8
Surgery tibia osteotomy group	110 (102–113)	100 (93–111)	0.7
Surgery intramedullary nail group	52 (49–61)	48 (45–54)	0.2

**Table 3 animals-11-02567-t003:** Median duration of loss of sensation at the coronary band, median duration of analgesia at surgical wound and time to standing were compared between groups (in hours). In group P 10/12, animals had no signs of sensation at the coronary band when the area of the wound had regained sensation completely while no animal ever showed a loss of sensation at the sacral area. * The effect duration was assumed based on literature [35] as in most sheep of group P sensation at the coronary band returned after the last evaluation. On the next morning, sensation had returned in all sheep.

	Peripheral Block	Epidural	*p*-Value
Duration of loss of sensation at coronary band (h)	10 (9.5–11.75) *	6 (5–8)	<0.001 *
Duration of analgesia at surgical wound (h)	6 (5–8.5)	8 (6.5–9)	0.17
Duration of loss of sensation in sacral area (h)	0	8 (6.3–8.8)	<0.001
Time to standing (h)	4 (3–4.5)	7 (6–7.5)	<0.001

**Table 4 animals-11-02567-t004:** Results of the mixed regression models with which VAS, SGS, general pain score and pain evaluation of the operated limb were compared between the groups P (peripheral block of the femoral and sciatic nerves) and E (epidural injection of ropivacaine). In all models, a group effect, a time effect and an interaction effect between group and time were evaluated.

	Estimate	Standard Error	*p*-Value
VAS			
Group (P vs. E)	0.40	0.42	0.34
Time	−0.10	0.04	0.04
Group*Time interaction	−0.11	0.07	0.11
SGS			
Group (P vs. E)	3.89	1.47	0.008
Time	−0.31	0.16	0.052
Group*Time interaction	−0.68	0.24	0.004
Duration of loss of sensation at coronary band			
Group (P vs. E)	−4.99	1.60	0.002
Time	−0.87	0.21	<0.001
Group*Time interaction	1.17	0.23	<0.001
Duration of analgesia at surgical wound			
Group (P vs. E)	−5.48	1.71	0.001
Time	−0.75	0.21	<0.001
Group*Time interaction	0.68	0.25	0.007

## Data Availability

The data presented in this study are available on request from the corresponding author.

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
