# Peer review of "Is a Block of the Femoral and Sciatic Nerves an Alternative to Epidural Analgesia in Sheep Undergoing Orthopaedic Hind Limb Surgery? A Prospective, Randomized, Double Blinded Experimental Trial"

_animals, 2021, doi:10.3390/ani11092567_

Round 1

Reviewer 1 Report

Thank you for the opportunity to review a very interesting and important study. I think it is important that the study be published but believe there are some changes to be made before this can be completed. In general, there is a lot of data to be presented and I think the methodology and results could do with refinement to better support the authors conclusions. 

Reviewer 2 Report

Thank-you for this submission, which is undoubtedly useful for anaesthetists dealing with research procedures in sheep, which are common. 

I have a few comments and queries. 

Line 30: Might be more correct to state ropivacaine/saline injected ‘ around ‘ or ‘over’ the nerve and ‘into’ the epidural space rather than ‘to’

Line 48: Systemically administered

Line 67: ‘technique is equivalent to a NS’ is not well expressed (the technique is perhaps “equivalent to that performed using a NS”)

Line 134: was 10 mL injected over each of the femoral and sciatic nerves (ie 20 ml total) or 10 ml injected over both nerves, ie 2 nerves each 5 mL

Line 174: You state that the fR was used as one determinant of lack of analgesia, however as you provided IPPV for all animals, I am not sure how this could be determined? Was intraoperative nociception really taken fro the HR and BP, and not fR? It might be more reasonable to just state the 2 variables which you could assess.

Line 247: This line states that one sheep in the epidural group had to be euthanased, however in the flow diagram above, it says one sheep in the peripheral nerve block group. Can you clarify and correct whichever site is in error please?

Line 250+: I couldn’t see anywhere how the different types of surgery were represented in the different groups (epidural vs block)- I know you specified you felt them to induce similar amounts of pain, but I do think we need to know how they were spread, were they equal between the 2 groups?

Line 259: I doubt you measured times accurately to less than 1 minute, so please can I suggest you describe the mean durations to the nearest minute

Line 305: I do not think you can really include an assumption for duration of action as a result you obtained and I think there needs to be a different way of expressing this information- perhaps just saying >6 hours if you couldn’t determine what it was for these animals in this study.

Round 2

Reviewer 1 Report

Thank-you for your consideration of the previous review comments and for the presenting the revised manuscript. 

I only had a couple of minor text edits to suggest:  

Abstract Line 36/37 - add P-values to explain significance of the results. 

Line 57: "Unfortunately, local anaesthetics lead not only to loss of sensation but loss of motor function as well."

Line 204: font for 'mg' is different/incorrect 

Line 299: "time they had gained sensation at the surgical wound they but could still not properly use the leg"

Line 363: "However, one drawback with the use of peripheral nerve blocks was the frequent occurrence ..."